# Development of a Micropropagation Protocol for the Ex Situ Conservation of Nuttall’s Scrub Oak (*Quercus dumosa*)

**DOI:** 10.3390/plants13081148

**Published:** 2024-04-20

**Authors:** Joseph Francis Ree, Christy Powell, Raquel Folgado, Valerie C. Pence, Christina Walters, Joyce Maschinski

**Affiliations:** 1San Diego Zoo Wildlife Alliance, San Diego, CA 92101, USA; 2Huntington Library, Art Museum, and Botanical Gardens, San Marino, CA 91108, USA; 3Center for Conservation and Research of Endangered Wildlife (CREW), Cincinnati Zoo & Botanical Garden, Cincinnati, OH 45220, USA; 4USDA—ARS National Laboratory for Genetic Resource Preservation, Fort Collins, CO 80521, USA; 5Center for Plant Conservation, Escondido, CA 92027, USA

**Keywords:** ex situ conservation, micropropagation, basal salts, antioxidants, in vitro germination

## Abstract

Worldwide, oak species are threatened with extinction due to habitat loss, pathogens, and changing fire regimes. Ex situ conservation through tissue culture may protect the remaining genetic diversity of *Quercus dumosa*, or the coastal sage scrub oak, from further loss. We designed three basal salt formulations based on the mineral composition of shoot tips and first leaves from mature *Q. dumosa* and explored carbohydrate source, stress-mitigating compounds, and plant growth regulator concentrations to develop a method of cultivating many *Q. dumosa* culture lines in vitro. All three novel basal salt formulations led to decreased necrosis compared with commercial basal salt formulas WPM, MS, and DKW. Substitution of 30 g L^−1^ sucrose with glucose and adding 250 mg L^−1^ ascorbic acid, 5.2 mg L^−1^ SNP sodium nitroprusside, and 103 mg L^−1^ y-aminobutyric acid improved culture health overall. In an experiment involving 115 culture lines, 0.66 mg L^−1^ 6-benzylaminopurine produced the highest average shoots per explant, but 0.33 mg L^−1^ produced the greatest proportion of shoots 2 cm or greater. Incubation for 24 h in 20 mg L^−1^ indole-3-butyric acid led to the most rooting. These methods show promise for the ex situ conservation of many genotypes of endangered *Q. dumosa*.

## 1. Introduction

With an estimated 31% of oak species under threat of extinction [1], developing tools to conserve them is paramount. Unfortunately, acorns are sensitive to drying, meaning they can only be stored for short periods before they lose viability, making seed banking unsuitable for ex situ conservation. Cryopreservation of oak embryonic axes is an alternative, but the tissue must recover post-cryopreservation on the tissue culture medium capable of cultivating a broad genetic diversity of the species. 

The coastal sage scrub oak, or Nuttall’s scrub oak (*Quercus dumosa*), is endemic to the coastal chaparral ecosystem of Southern California and Baja California. For centuries, the native peoples managed the land by burning areas to clear duff and creating open areas that promoted the growth of a wide range of species during post-fire recovery [2]. Oaks were favored for their acorns and building material [3]. However, agricultural and urban expansion in the last century brought severe threats to the coastal chaparral ecosystem through land clearing and increased fire frequency caused by more sources of ignition and the introduction of highly flammable non-native grasses [4]. As a result, *Q. dumosa* is listed on the IUCN red list as endangered [5]. Conserving this species is essential for the coastal chaparral ecosystem’s well-being and preserving the living history of the first peoples who shaped it.

Tissue culture for ex situ conservation allows genetic diversity to be stored away from wildfire and disease. Mass multiplication of a single culture line does not serve conservation because all resulting plants are genetically identical; therefore, culture lines must be initiated from enough wild individuals to capture genetic diversity [6]. However, every genotype responds to growing conditions differently; therefore, any possible improvement to a protocol must be based on a ‘consensus’ based on overall improved growth across a wide range of tested individuals. For example, shoots of three pear species (*Pyrus communis* (three cultivars), *P. dimorphophylla*, and *P. ussuriensis*) grown on 1 of 43 different combinations of basal salt components showed that each genotype grew differently from the others, but the authors were able to identify several overall results: all three showed decreased abnormal growth with greater concentrations of mesos (CaCl_2_, KH_2_PO_4_, and MgSO_4_), and tissue necrosis and abnormal growth generally increased with low mesos and nitrogen [7]. By identifying the common factors that improve growth within a species, it may be possible to cultivate many individuals in vitro.

Commercial basal salt formulas developed for specific plants, such as woody plant medium (WPM) developed for mountain laurel (*Kalmia latifolia*) [8], are readily available. Although many species grow well in these basal salts, there is great interspecific and intra-specific variation in performance on the media [9,10,11]. Several studies found successful methods of calculating new basal salt formulations based on the mineral analysis of field tissue, though each method is distinctly different [12,13,14,15,16]. Monteiro et al. [12] found that passionfruit (*Passiflora edulis*) grown in vitro showed frequent leaf bleaching and reduced growth. They used leaf mineral analysis and data from the available literature to design a new medium. This new medium contained, among other changes, increased Fe and Ca, resulting in healthier plants without the visual symptoms of mineral deficiency. Terrer and Tomás [14] used leaf mineral analysis from specific peach–almond hybrids (*Prunus persica* × *P. amygdalus*) to create custom basal salt mixtures tailored specifically to each hybrid with the result that they showed greater growth compared with the widely used Murashige and Skoog [17] (MS) basal salts. Staikidou et al. [16] kept the N composition of MS, but modified all other inorganic elements based on the ratio of N to other inorganic elements found in a mineral analysis of *Galanthus* bulblets. Similarly, Lozzi et al. [15] used the mineral analysis of carob (*Ceratonia siliqua*) cotyledons to design a culture medium. These authors used a fixed concentration of nitrogen known to be non-toxic to carob and modified the concentration of each other component of the culture medium based on the ratio of nitrogen to other elements found in the cotyledons. The resulting medium increased carob shoot height and multiplication rate while decreasing shoot tip necrosis.

In addition to their mineral requirements, tissue cultures may require additives to provide conditions suitable for preventing stress and promoting growth. For example, shoot tips harvested from 12 oak species placed in tissue culture exuded considerable amounts of phenolic substances into the medium, requiring frequent medium changes [11]. Nevertheless, most of the tissue in this study died within less than a month, and only 5 explants, representing 2 of the 12 studied species, from a total of 216 noncontaminated explants showed shoot growth. However, exogenous antioxidants reduced signs of stress in the oaks *Q. robur* and *Q. petraea* [18], preventing rapid tissue necrosis and promoting growth.

Our goal was to develop a culture medium capable of growing a wide range of *Q. dumosa* culture lines by modifying a ‘standard’ medium into one capable of growing many culture lines. Specifically, we aimed to design three new basal salt formulations based on three chosen studies and explore other ways to create culture conditions suitable to *Q. dumosa*, including carbohydrate source, the stress-mitigating compounds ascorbic acid (AA), activated charcoal (AC), γ-aminobutyric acid (GABA), and sodium nitroprusside (SNP), and the concentration of the growth regulators 6-benzylaminopurine (BAP) and gibberellic acid (GA_3_).

## 2. Results and Discussion

### 2.1. Germination and Line Establishment

Of the 360 acorns harvested from wild trees (Figure 1a) placed in vitro in 2020, 67 became contaminated (18.6%). Of the remaining 293, 213 germinated (72.7%) (Figure 1b). Each of these germinated plants were sectioned to create culture lines; however, only 157 of the 213 culture lines showed capacity to resume growth and multiply, with the remaining 56 showing rapid necrosis, exudation of phenolic compounds (Figure 1c), and death within a few days to a month. Only 11 lines showed reliable growth, with shoot numbers consistently increasing after half a year with monthly subcultures to establish a population of shoots (Figure 1d). All others were non-responsive, or their growth was sufficient only to replace shoots lost to necrosis (Figure 1e). Of the 157 lines established in 2020, only 52 survived through the end of this series of experiments in 2023, with most lost due to culture decline due to the inability to fully establish in vitro. Of the 108 acorns placed in vitro in 2021, 36 became contaminated (33.3%), and 66 of the remaining 72 *Q. dumosa* embryos germinated (91.7%). Of these, 60 successfully established culture lines, and 49 survived through the end of this series of experiments in 2023. The ability of each culture line to grow existing meristems varied widely, with many remaining senescent for weeks or months without any apparent change. This seeming episodic growth is common in other oaks [19]. Others died quickly through the exudation of phenolics and necrosis. The combination of slow growth, genotypic dependence, and necrosis through exudation of phenolic compounds extends to other white oaks [20,21]. 

### 2.2. Development of a New Basal Medium

Data from the mineral analysis of wild *Q. dumosa* shoot tips and first leaves (Table 1) showed detectable amounts of N, B, Ca, Cu, S, Fe, Mg, Mn, K, P, and Zn as a % of the total mass of the sample. We used three methods [12,13,14,16] to formulate three basal salt formulas based on the three methods (Table 2). Notably, the mineral analysis only detected total N, but not whether it was in the form of NH_4_^+^ or NO_3_^−^. All three novel formulas included both, but the ratio between either form of nitrogen may be important for the healthy growth of cultures, as shown in *Cocos nucifera* [22]. Optimizing the ratio between NH_4_^+^ and NO_3_^−^ may be necessary in future work.

Overcoming necrosis is one of the most critical steps to resolve for conservation tissue culture, especially when working with endangered species, as the material obtained from wild populations is limited in abundance and may be irreplaceable over time. Culture lines must show low necrosis to prevent the need to resample an individual if the culture line dies. Necrosis was highest in shoot and nodal segment explants cultured on MS (45.2 ± 5.5% and 56.9 ± 8.7%, respectively) or DKW [23,24] medium (41.8 ± 7.1% and 55.1 ± 6.2%, respectively) (Table 3). In shoot tip explants, the remaining four basal salt types were not significantly different, although WPM showed 15.6 ± 3.2% necrosis, compared with 2.1 ± 1.6% for Qd1, 4.0 ± 1.8% for Qd2, and 1.9 ± 1.3% for Qd3. Necrosis showed a similar pattern in nodal segment explants. Qd1 (20.5 ± 6.0%), Qd2 (17.7 ± 7.1%), and Qd3 (21.2 ± 6.6%) showed less necrosis than WPM (39.6 ± 7.3%). Reasons for necrosis were not determined and would require further experimentation; however, Teixeira da Silva et al. [25] reviewed possible factors contributing specifically to shoot tip necrosis in vitro: Ca deficiency; B deficiency; N type, quantity, and ratio between ammonium or nitrate; subculture length; basal salt choice; type, quantity, and interactions between exogenous plant growth regulators; ethylene accumulation within the vessel; lack of exogenous antioxidants; genotype-related factors; and other factors. Teixeira da Silva et al. [25] also highlighted that adding more of one ion, such as Ca^2+^, also means that other ions are also added, such as either Cl^-^ (CaCl_2_) or NO_3_^−^, which can confound the interpretation of the data. We can, therefore, only state that necrosis decreased on any of the three custom *Q. dumosa* basal salt types but not pinpoint specific mineral quantities as the reason why. All three do have several similarities, which can be evaluated in follow-up experiments. In comparison with commercially available media, the three novel Qd basal salt media had substantially greater amounts of Cu and lower Fe and Zn concentrations (Table 2). Interestingly, a study aimed at evaluating downy oak (*Q. pubescens*) for phytoremediation of soil contaminated with Cu or Cd found that the addition of 5 µM Cu increased growth, although higher concentrations decreased it [26]. Both DKW and MS contained greater quantities of total N than all three Qd salts or WPM, which may have been one of the major factors in why either MS or DKW had much greater necrosis than the other four. Additionally, MS and DKW both contained micronutrients that were not present in the other formulations: Co and Ni, respectively. Whether or not these elements contributed to the high necrosis in MS or DKW is unknown.

Of the six basal salt mixtures tested, WPM and the three novel Qd mixtures showed comparable growth rates of new shoots. WPM showed comparable formation of shoots at least 0.5 cm in height (1.0 ± 0.1 per explant) and a similar percentage of those shoots reaching 1 cm or greater (36.6 ± 4.6%) compared to Qd1 medium (1.1 ± 0.1 and 41.2 ± 6.2%, respectively) in shoot tip explants. Despite having the lowest necrosis of the three novel mixtures, shoot tip explants on Qd3 had the least growth of shoots at least 0.5 cm in height (0.9 ± 0.1) and proportion of shoots reaching lengths 1 cm or greater (21.3 ± 8.0%). Shoot explants on Qd2 had intermediate values to the other Qd mixes, with 0.9 ± 0.2 shoots 0.5 cm or greater per explant and 28.8 ± 9.1% of those greater than 1 cm. The two most effective basal salts for generating shoots on nodal segment explants at least 0.5 cm in height were WPM (0.8 ± 0.1 per explant) and Qd1 (0.6 ± 0.2 per explant). The number of shoots produced across all six treatments for the 11 tested culture lines using nodal segment explants was insufficient to justify continued use for future experiments.

In addition to the high rates of necrosis, shoot tips placed on a medium with MS or DKW likewise showed a statistically significant decrease in the number of shoots per explant compared with those on WPM, Qd1, or Qd2 according to a Tukey post hoc test (*p* < 0.05). There was no significant difference between WPM or any of the Qd basal salts for either the number of shoots per explant for shoot tip or nodal segment explants, nor was there a difference in the percentage of shoots greater than 1 cm. None of the Qd basal salts led to increased growth compared with WPM; however, the quality of the shoots showed some qualitative differences that are hard to quantify accurately. Shoots grown on WPM medium tended to show frequent leaf discoloration, tissue darkening, and lesions on the leaves (Figure 1f), which were less frequent when cultivated on any of the Qd mixtures (Figure 1g). Considering the deceased necrosis and comparable growth rates, Qd1 medium replaced WPM as the basal salt mixture for *Q. dumosa* for subsequent trials of carbohydrate source and additives (Figure 2). Qd1 was chosen over the other two *Q. dumosa* basal salts because it showed slightly greater growth overall. These results support the effectiveness of the three methods used to design novel basal salts [12,13,14,16]. Other methods exist and have shown effectiveness for other species. For example, Rugini [27] developed a medium for olive (*Olea europaea*) based on mineral analysis of mature embryos and apical shoots, finding that multiple cultivars showed a similar increase in culture health and vitality. As a result of replacing WPM with Qd1, an additional four *Q. dumosa* culture lines produced sufficient shoots to be included in further experiments.

### 2.3. The Effect of Carbohydrate Source

Shoots on glucose showed the least overall necrosis (1.8 ± 3.2%, *p* < 0.05) compared with sucrose (9.6 ± 1.3%) and mannitol (64.1 ± 10.7%) (Table 4). To establish a baseline for evaluating changes in *Q. dumosa* culture medium, the eleven core lines and the four additional lines were also examined separately as well as pooled. Among the core 11 culture lines, necrosis in sucrose (4.4 ± 2.7%) and glucose (0.4 ± 0.7%) were not significantly different. The four additional culture lines showed 24.0 ± 6.7% necrosis in sucrose and decreased necrosis in glucose (5.6 ± 3.9%). Mannitol trials had complete necrosis of all shoot tips from the four additional culture lines and 51.0 ± 12.6% of the core eleven culture lines. This trend carried over into the average number of shoots per explant. Mannitol significantly diminished overall shoots per explant (0.4 ± 0.1, *p* < 0.05) compared with glucose (1.9 ± 0.2) or sucrose (1.6 ± 0.2). Mannitol nearly inhibited all growth in the core 11 culture lines with 0.5 ± 0.1 shoots per explant. By contrast, the core 11 culture lines produced 1.7 ± 0.2 shoots per explant on sucrose and 2.1 ± 0.1 shoots on glucose. The additional lines showed comparatively fewer shoots per explant, with 1.2 ± 0.1 shoots on sucrose and 1.4 ± 0.1 shoots on glucose. These results contrast with the higher shoot length, the number of shoots, and leaf formation seen in olive (*Olea europaea*), another woody plant native to a Mediterranean-type climate, when cultured on mannitol compared with sucrose [28]. Valonia oak (*Q. aegilops*) grew well on either glucose or sucrose; however, the concentration of either also mattered [29]. Sucrose at 30 g L^−1^ and glucose at 45 g L^−1^ showed comparable multiplication rates. Because of the decreased necrosis and the greater numbers of shoots per explant, glucose was substituted for sucrose as the carbohydrate in the *Q. dumosa* culture medium. *Quercus robur* formed roots most effectively in a medium with 65 g L^−1^ glucose compared with lower concentrations [30]. Future work may analyze different concentrations of glucose on shoot growth and health. Additionally, these differences in concentration might help rooting. As a result of replacing sucrose with glucose, an additional five culture lines began to produce enough shoots to be included in further experiments, raising the total number of additional culture lines to nine.

### 2.4. The Effect of Activated Charcoal and Ascorbic Acid

Exogenous AA or AC significantly decreased overall necrosis in *Q. dumosa* (0.0 ± 0.0% and 1.3 ± 0.9%, respectively, *p* < 0.05) compared with the medium without either additive (9.5 ± 3.4%) (Table 5). None of the shoot tips in the core 11 culture line showed necrosis when cultivated with 5 g L^−1^ AC, 250 mg L^−1^ AA, or without either compound. Only 2.8 ± 2.1% of shoot tips from the nine additional lines showed necrosis when cultivated with 250 mg L^−1^ AA, compared with 21.3 ± 5.6% without either antioxidant. Although AC eliminated necrosis rates among all 20 tested culture lines, it also significantly decreased the production of shoots per explant (1.1 ± 0.1, *p* < 0.05) compared with the medium without antioxidants (1.6 ± 0.1) or with AA (1.7 ± 0.1). The 11 core culture lines showed minimal shoots per explant (1.0 ± 0.1) when cultivated with AC, and the additional lines showed no growth at all. Although AA decreased necrosis, it did not lead to overall increased shoots per explant. In the core lines, 1.8 ± 0.2 shoots grew without AA compared with 1.9 ± 0.2. The additional lines showed a small increase from 1.2 ± 0.1 to 1.4 ± 0.1 shoots per explant. 

These results are similar to those of lavandin (*Lavandula x intermedia*) in which both AA or AC improved culture health and ability to grow [31]. However, that paper used 1 g L^−1^ AC, whereas the medium used with *Q. dumosa* contained 5 g L^−1^. The addition of 3 g L^−1^ AC similarly led to decreased numbers of shoots formed per explant and increased the rate of hyperhydricity in Japanese pear (*Pyrus pyrifolia*) [32]. No hyperhydricity was evident in any of the 20 culture lines during the experiment, but, at the tested concentration, AC showed a distinct tradeoff: high survival but near cessation of growth. Although AC is not overall useful for multiplying cultures, it may be useful for long-term storage of culture lines that need to be maintained but not expanded. The addition of 250 mg L^−1^ AA decreased necrosis in *Q. dumosa* is similar to the results found in the aster *Brahylaena huillensis*, in which 200–250 mg L^−1^ controlled the production of phenolic compounds [33]. Although not as effective in controlling necrosis as AC, 250 mg L^−1^ AA improved overall culture health and became a standard additive to *Q. dumosa* medium. As a result, an additional seven culture lines began to produce enough shoots for further experiments, raising the total number of additional culture lines to 16.

### 2.5. The Effect of Sodium Nitroprusside

Both 5.2 and 10.5 mg L^−1^ SNP significantly decreased necrosis (1.0 ± 1.0% both, *p* < 0.05) compared with the medium without SNP (7.3 ± 2.4%). (Table 6). Notably, necrosis was higher in the thirteen additional lines (11.5 ± 3.6%) than in the nine in the previous study (2.8 ± 2.1%). This is likely due to the inclusion of four new sensitive culture lines and simple variation between experiments. For example, no necrosis occurred in the core 11 culture lines during the antioxidant study but did so in this study (2.6 ± 2.3%). The addition of 5.2 mg L^−1^ SNP significantly increased the number of shoots per explant (2.0 ± 0.1) compared with the medium without SNP (1.8 ± 0.1). There was no apparent trend among the core lines in response to SNP, but the additional lines showed increased shoots per explant at concentrations beyond 1.3 mg L^−1^ SNP, with the greatest number of shoots per explant at 5.2 mg L^−1^ (1.8 ± 0.1) compared with those cultivated without SNP (1.5 ± 0.1), although this increase was not statistically significant. Some culture lines often responded strongly to different concentrations, while others were unaffected. For example, the two core lines that consistently grew the most shoots per explant showed decreased growth with SNP, whereas another core line showed considerable growth. The chemical decomposition of SNP releases NO gas, which affects many cell processes, including those involved with both abiotic stress and cell signaling through ethylene, jasmonic acid, and salicylic acid (reviewed in Hasanuzzaman et al. [34]). Persian shallot (*Allium hirtifolium*) treated with SNP showed increased protection from PEG-related osmotic stress, decreased hydrogen peroxide accumulation, and cell membrane peroxidation with a corresponding increase in superoxide dismutase and ascorbate peroxidase enzyme activity [35]. Soybean (*Glycine max*) [36] and *Ficus religiosa* [37] showed similar results. Tan et al. [38] found that 5–15 µM SNP increased vanilla (*Vanilla planifolia*) mean shoot number; however, higher concentrations were detrimental. More research is needed to verify to what extent SNP affects cellular activity in *Q. dumosa*. However, 5.2 mg L^−1^ SNP had an overall positive effect in the additional lines, as shown by the trend of decreased necrosis and an increase in average shoots per explant.

### 2.6. The Effect of γ-Aminobutyric Acid

There was no significant difference in necrosis between the medium without GABA (2.1 ± 1.2, *p* = 0.53) and medium with 51.5 mg L^−1^ GABA (3.5 ± 1.7%) or 103 mg L^−1^ GABA (1.4 ± 1.0%) (Table 7). Among both the 11 core and 13 additional lines, there was no trend with necrosis. However, 103 mg L^−1^ GABA did significantly increase the number of shoots per explant (2.2 ± 0.1) compared with the medium without GABA (2.0 ± 0.1) or with 51.5 mg L^−1^ GABA (2.0 ± 0.1). No trend was evident in the 11 core culture lines, but the 13 additional lines showed a slight, although not significant, uptrend of a greater number of shoots per explant from 1.5 ± 0.1 to 1.6 ± 0.1 (mg L^−1^ GABA) to 1.8 ± 0.1 (103 mg L^−1^ GABA). GABA plays a number of possible roles in plant biochemistry, including metabolism, biotic and abiotic stress signaling, and directing tissue growth patterns (reviewed in Seifikalhor et al. [39]); however, much is still unknown. Fresh-cut potato (*Solanum tuberosum*) tubers treated with 20 mM GABA for 10 min showed decreased tissue browning due to increased catalase and superoxide dismutase activity [40]. Exogenous GABA improved pineapple guava (*Acca sellowiana*) somatic embryo formation by improving normal formation while decreasing the number of malformed somatic embryos [41]. For *Q. dumosa* culture lines, GABA had either no overall effect or a slight positive effect on the average number of shoots per explant. Both 103 mg L^−1^ GABA and 5.2 mg L^−1^ SNP were added to version 5 medium to create the final medium of this work (Figure 2).

### 2.7. The Effect of 6-Benzylaminopurine and Gibberellic Acid

GA_3_ had no significant effect on necrosis (*p* = 0.75), average shoots per explant (*p* = 0.48), or production of shoots greater than 2 cm (*p* = 0.51). This was surprising because GA_3_ is frequently used to stimulate shoot elongation, such as in Guava (*Psidium guaja*) [42] and Moru oak (*Quercus floribunda*) [43]. Therefore, we ignored GA_3_ as a factor and considered only BAP concentration among the 11 core, 16 additional, and 88 other lines, termed as ‘static’ lines, because their populations of shoots remained consistently low either due to necrosis or slow growth (Table 8). Omitting BAP led to significantly increased necrosis (25.1 ± 4.1%, *p* < 0.05) across all lines compared with any amount of BAP. This was especially true for static lines cultivated without BAP (37.7 ± 5.9% necrosis). Although necrosis decreased with BAP, the lowest necrosis for static lines was 0.66 mg L^−1^ (15.1 ± 4.4%). The 11 core and 16 additional lines showed their highest necrosis when cultivated without BAP (7.5 ± 2.5 and 9.6 ± 3.9%, respectively). No necrosis occurred in the core lines with any concentration of BAP, and the 16 additional lines showed necrosis below 5% at any BAP concentration. 

Similar to the results for necrosis, the lack of BAP was highly detrimental to growth among the three types of culture lines, with the static lines, in particular, showing fewer viable shoots at the end of the experiment than when it started (0.7 ± 0.1 average shoots per explant, *p* < 0.05). Oleander (*Nerium oleander*) shoot tip cultures showed a similar loss in culture health when cultivated without exogenous cytokinin [44]. *Rhododendron* shoots showed the least growth when cultured without zeatin, but the lack of cytokinin did not wholly inhibit growth [45]. Several of the core and additional lines had positive growth even without BAP but less than any of the tested BAP concentrations. All three culture lines showed a faint trend: 0.66 mg L^−1^ BAP promoted the greatest number of shoots per explant for each culture type: 2.6 ± 0.2, 1.9 ± 0.2, and 1.8 ± 0.2 for core, additional, and static lines, respectively. All culture lines had the second most growth at 1 mg L^−1^ BAP (2.6 ± 0.2, 1.8 ± 0.2, and 1.6 ± 0.2) and third most with 0.33 mg L^−1^ (2.3 ± 0.2, 1.6 ± 0.2, and 1.6 ± 0.1). The optimal BAP concentration differed among four species of Mexican oaks, but the general trend showed that the lack of any exogenous cytokinin was detrimental to shoot proliferation [46]. *Q. dumosa* is similar in this respect: the optimal BAP concentration differs between culture lines, but no culture line grew well without BAP. 

BAP significantly affected the growth of shoots 2 cm or greater (*p* < 0.05). The lack of BAP led to the least growth of tall shoots (6.3 ± 9.1, 6.5 ± 6.5, and 2.2 ± 4.3% for core, additional, and static lines, respectively). Conversely, 0.33 mg L^−1^ yielded the highest percentage of shoots with lengths 2 cm or greater for core and additional lines (49.4 ± 15.0 and 23.7 ± 8.7%). Static lines showed the highest proportion of shoots 2 cm or greater when cultured on 0.66 mg L^−1^ (15.4 ± 5.7). Moru oak (*Q. floribunda*) shoots proliferated more as BAP concentration rose from 0 to 22.19 µM (~5 mg L^−1^) [43], and higher cytokinin concentrations also led to greater numbers of guava shoots [47]. Concentrations greater than 1 g L^−1^ BAP might lead to a similar proliferation of *Q. dumosa* shoots. However, the highest concentration attempted here did not have the greatest number of shoots for the three culture line types. The BAP concentration used for *Q. dumosa* medium can be changed based on what is required: 0.33 mg L^−1^ BAP to stimulate shoot elongation and 0.66 mg L^−1^ BAP for shoot proliferation.

### 2.8. Comparison between Initial and Final Medium Compositions

The initial medium composed of WPM salts and vitamins, 30 g L^−1^ sucrose, 1 mg L^−1^ BAP, and 8 g L^−1^ agar led to higher necrosis (*p* < 0.05) and fewer average shoots per explant (*p* < 0.05) across a total of 25 culture lines compared with the final medium composition (Table 9). Overall, 51.0 ± 6.9% of shoot cultures became necrotic across a combined 11 core and 14 additional lines when placed on the initial medium compared with 6.5 ± 1.9% on the final medium. The core lines showed less necrosis on the initial medium (29.5 ± 8.1%) than the additional lines (67.9 ± 7.8%). The same was true for core lines on the final medium (1.1 ± 1.1%) compared with additional lines (10.7 ± 2.8%). Five of the fourteen lines showed complete necrosis on the initial medium, but none did so on the final medium. Curiously, necrosis among the core lines was nearly double that of the basal study test (15.6 ± 3.2% vs. 29.5 ± 8.1%). There may be multiple reasons for this increase in necrosis. Moving cultures from an improved medium to one known to cause stress in cultures may have caused an abiotic shock. These experiments took place over several years and many subculture cycles, which may also have had an effect. Over time and many subcultures, plant cells may acquire epigenetic changes [48], resulting in a gradual decline in health. The increase in necrosis found in the core 11 lines might be a symptom of culture decline because the same culture lines have existed over multiple years. For conservation, the gradual decline of plant tissue cultures poses a problem; however, storing cultures at cooler temperatures or keeping them in conditions that will slow their growth, such as placing *Q. dumosa* shoots on a medium with activated charcoal, might increase the time a culture line is viable for ex situ conservation. 

The final medium produced significantly more shoots per culture line 1.87 ± 0.23 compared with the initial medium (0.86 ± 0.15) (*p* < 0.05). The core lines produced 1.27 ± 0.25 shoots per culture on the initial medium compared with 2.41 ± 0.43 on the final medium, and additional lines produced 0.53 ± 0.14 and 1.45 ± 0.14, respectively. However, it is important to note that, of the fourteen additional lines, three neither grew nor showed signs of stress, such as discoloration, shoot tip necrosis, or chlorosis. Instead, they remained unchanged for the whole experiment. The other 22 culture lines showed varied growth depending on the culture line, with some multiplying slowly and others producing many shoots. Large numbers of shoots are not strictly necessary for ex situ conservation, but consistent growth ensures that a culture line can remain stable and produce new whole plants without risk. The final medium composition overall succeeded compared with the initial medium.

### 2.9. Rooting

A 24-hour pulse with 20 mg L^−1^ IBA transferred to a medium devoid of growth regulators resulted in eight culture lines (with AC) or seven (without AC) culture lines with at least one rooted shoot (Table 10) (Figure 1h). This method was more effective than prolonged incubation for 30 days in 5 mg L^−1^ IBA, which produced roots in three or five culture lines with or without AC, respectively. No shoot tips in any of the culture lines rooted via the 15-second dip in the 1 mg ml^−1^ IBA method. Between the two other methods, the 24-hour pulse in 20 mg L^−1^ IBA produced a higher proportion of rooted shoots (*p* < 0.05) both with AC (47.2 ± 10.2%) or without it (38.9 ± 10.1%) across the nine tested culture lines. Prolonged incubation resulted in 4.2 ± 2.1% or 16.7 ± 6.9% rooting with or without AC, respectively. Rooting depended heavily on culture line. For example, the eight culture lines subjected to the 24-hour pulse method with AC showed between 25–100% rooting, and the seven culture lines in the same method without AC had 12.5–75% rooting. Different culture lines of a plant might require specific conditions to form roots. For example, Reed [49] required multiple experiments to find conditions to root all but 11 of the tested 49 species and cultivars of Pyrus and an accession of *Pyronia veitchii*. Similarly, rooting success of pedunculate oak depended on specific clones, which showed rooting capacity between 15 and 46% [50]. Sanchez et al. [51] found that rooting success depended on genotype and method: several genotypes required a pulse of high concentrations of IBA for a day, others responded more to a short dip treatment, and a single genotype failed to root unless it was on a medium with AC. Four oak species (*Q. rugosa*, *Q. resinosa*, *Q. eduardii*, and *Q. castanea*) each showed different abilities to root, but placing shoots on a medium with 10 mg L^−1^ IBA for 2 or 5 days each produced the highest overall rooting percentages [46]. Considering that *Q. dumosa* showed greater rooting with a similar protocol, the pulse method is worth exploring more for rooting a greater number of culture lines. More concerning, however, is that *Q. dumosa* either grew slowly after rooting or ceased growth entirely. They became senescent, similar to culture lines placed on AC for long-term storage. Resuming growth after rooting is essential for acclimation and return of plants from agar to soil. Oaks in the dry environment of California rely on a complex symbiotic relationship with mycorrhizal fungus to survive drought conditions (reviewed in Allen [52]). The lack of mycorrhizae could be one reason why the *Q. dumosa* rooted plantlets showed little to no growth. Rooted *Castanea sativa* plantlets inoculated with the mycorrhizae *Pisolithus tinctorius* showed greater shoot height and diameter and overall greater dry mass compared with non-inoculated plantlets [53]. Further, *Quercus suber* plantlets exposed to several species of mycorrhizal fungi showed greater secondary root growth and greater acclimatization when plantlets were transferred to ex vitro conditions [54]. Considering the low growth after rooting, the absence of mycorrhizae that help oaks like *Q. dumosa* survive in their natural habitat [52], and the greater growth of related species upon exposure to mycorrhizae [53,54], mycorrhization of rooted *Q. dumosa* should be investigated. That investigation is now possible with a reliable means of producing healthy *Q. dumosa* shoots from an improved tissue culture medium.

### 2.10. Considerations for Working with Slow-Growing and Largely Unstudied Species for Conservation

The tissue culture of endangered species possesses several intrinsic obstacles: the medium must suit many individuals of many populations; the most directly applicable literature is often limited to other members of the same family instead of the species itself; and it might be challenging to generate enough usable tissue for large experiments if the plant grows slowly or is easily stressed. The latter was a significant hindrance in earlier experiments because many culture lines often showed a net loss of shoot tips during multiplication due to necrosis and signs of stress. Any component of the tissue culture medium can affect plant health, so one must approach the problem by considering each one individually.

Basal salts are often considered a single testable factor in tissue culture experiments but are mixtures of many elements tailored for the specific requirements of a single species. For example, MS was developed for tobacco (*Nicotiana glauca × langsdorffii*) [17], a herbaceous Solanacaea, and WPM was developed for Mountain Laurel (*Kalmia latifolia*) [8], which, though a woody shrub, grows far away from the native range of *Q. dumosa*. Modifying these mixtures is labor intensive and requires considerable available tissue, which may preclude slow-growing or endangered species from study because there is either insufficient tissue available or gathering that tissue might be destructive. By finding the mineral composition of growing apices, one only needs to test several methods of interpreting that data [12,13,15,16,27] and translating it into a basal salt mixture. We tested three, and all three led to decreased necrosis, which allowed the generation of more tissue from more culture lines. We do not have the data to pinpoint what components of the three were the most important. Further investigations, such as modifying the ratio of NH_4_^+^ to NO_3_^−^ might lead to further improvements [22] but would still require large quantities of previous shoots. Many other factors can be tested for far fewer and may have greater effects on culture health. To develop a new medium based on mineral analysis of apical meristems and leaves, one must also consider how destructive it could be because a large amount of fresh tissue is required for mineral analysis. *Q. dumosa* is endangered, but the remaining individuals can grow large and bushy, allowing for tissue collection without inflicting significant damage. The same mineral analysis would be more destructive to rare species that might be fewer in number or produce less tissue. However, a basal medium created for a species within a region might be the best starting point for such a rare species that occurs alongside it (unpublished data). 

Beyond basal salts, some testable factors can be highly destructive to available culture stocks, as shown by the high necrosis of placing cultures on a medium with mannitol. The loss of many shoots delays further tests because those numbers must be recovered through additional multiplication cycles. *Q. dumosa* grows slowly in vitro, so these significant losses in tissue represent a considerable delay in protocol development. Preliminary studies with prolific culture lines can screen for factors worth further study and avoid those that might be damaging. Additionally, optimization, though useful for specific genotypes and elite cultivars, is not the end goal for the use of tissue culture for conservation. Each individual likely has its own ideal growing conditions, but finding those very specific conditions is impractical. It is sufficient to find conditions that are ‘good enough’ for all. Revisiting already-studied factors might bring further benefits but at the risk of diminishing returns. One must evaluate if those resources could be directed to new factors that might have an overall beneficial effect. However, every improvement, no matter how minor, may have a far-reaching impact on the ability to preserve a species with tissue culture.

## 3. Materials and Methods

### 3.1. Plant Material

Acorns were harvested from multiple populations of wild *Q. dumosa* within San Diego County, California, United States, in 2020 and 2021 to create the culture lines for all experiments (Figure 2). The pericarp of each acorn was cut superficially to create weak points, which could then be torn to free the embryo without damaging it. Whole embryos were then disinfected in 1.5% sodium hypochlorite for 15 min on an orbital shaker. The acorns were then transferred to a laminar flow hood, washed once with autoclaved water, and then placed on sterile paper to remove excess water. The testa was removed, and the embryo was then placed on the initial medium composed of WPM basal salts and vitamins, 30 g L^−1^ sucrose (Sigma-Aldrich, St. Louis, MO, USA), 1 mg L^−1^ 6-benzylaminopurine (BAP) (Phytotech Labs, Lenexa, KS, USA), and 8 g L^−1^ agar (Phytotech). The excised embryos were cultivated under a 16:8 light–dark photoperiod under fluorescent lights at 21 ± 2 °C. This initial medium served as the baseline medium for all subsequent media. The pH of all media was adjusted to 5.8 prior to autoclaving. AA, SNP, GA, and GABA were added by filter sterilization after autoclaving. In 2020, 360 *Q. dumosa* embryos were placed in vitro and a further 108 were added in 2021. Shoot tips and stem segments were sectioned from individual germinating embryos and placed onto fresh medium. Each medium vessel contained tissue from only a single individual, and the resulting shoots became culture lines. Each culture line was multiplied by micropropagation every 80 days. 

### 3.2. Development of a New Basal Medium

Fresh vegetative material consisting of shoot tips and emerging leaves was harvested from wild trees from three populations within San Diego County, California, USA: Torrey Pines State Reserve, Tecolote Canyon, and Los Peñasquitos Canyon Preserve. Approximately 0.1–0.3 g was taken from a single tree until 18 g were amassed from each of the three populations. These samples were pooled and sent to Waypoint Analytical (Anaheim, CA, USA) to determine the percentages of N, P, K, B, Ca, Mg, Mn, Zn, Cu, S, and Fe in the tissues (Table 1). With this data, three new basal salt formulations for *Q. dumosa*, referred to as Qd1, Qd2, and Qd3 (Table 2), were developed.

Qd1 salts were formulated based on an equation found in Correia [13], who calculated that 15 g of *Eucalyptus* spp. dry matter should result from growing after 21 d with 1 L of medium: (1)$$E_{medium}=\frac{E_{leaf}*\;15}{1000}$$

Equation (1) shows this formula, where *E_medium_* equals the concentration of a specific inorganic element in the medium (g L^−1^) and *E_leaf_* equals the concentration of the same inorganic element in shoot tips and leaves (g kg^−1^). For example, *Q. dumosa* shoot tips and first leaves were 0.45% calcium (Table 1), which would then be converted to 4.5 g kg^−1^. According to the equation, this would mean that *EM_Ca_* = 0.0675 g L^−1^. The type of calcium-containing salt must be considered, because the addition of CaCl_2_ or Ca(NO_3_)_2_ would further change the relative concentration of chlorine or nitrogen, respectively. Therefore, performing a further conversion to moles was necessary. The moles of each inorganic element were then used as a ‘target’ for calculating how much of a specific basal salt would be required using a spreadsheet. This was done by listing commonly used basal salts (See Table 2) and their molecular weights, which would then be used to calculate how many moles of each inorganic element would result from adding a given amount of a salt until within ±2% of the target amount of each element. 

Qd2 salts were formulated based on two equations in Terrer and Tomás [14], which use an already established basal salt medium as a reference for total macronutrient concentration. Terrer and Tomás [14] used MS medium as the reference for peach–almond hybrids, but we used WPM as the reference for *Q. dumosa*:(2)$$\alpha =\frac{\Sigma reference\;medium\;nutrients}{\Sigma leaf\;nutrients}$$
where *α* equals a multiplication factor used in a second equation:(3)$$E_{leaf\;}*\;\alpha =E_{formula}$$

The total moles into mg of each element found in 1 L of WPM was summed together for the numerator in Equation (2). The denominator of Equation (2) was the sum % of all inorganic nutrients found in *Q. dumosa* tissue. Equation (2) resulted in *α* = 326.7. The α value was then used in Equation (3) to find the total moles of each inorganic element required, which was then used to formulate the basal salt.

Qd3 salts were formulated based on the approaches of Staikidou et al. [16], who formulated a new basal salt formula based on a mineral analysis of *Galanthus nivalis* and *G. elwesii* bulbs. The authors kept the same N concentration as found in MS medium in the new medium, but modified all other inorganic elements based on a ratio between the N found in the bulbs and all other inorganic elements. We modified this approach by using the N concentration found in WPM: (4)$$E_{medium\;}=\frac{E_{leaf}}{E_{leafN}}*E_{mediumN}$$
where *E_medium_* is the moles of specific inorganic elements, *E_leaf_* is the % of the same inorganic element found in *Q. dumosa* shoot tips and first leaves, *E_leafN_* is the % of N in *Q. dumosa* shoot tips and first leaves, and *E_mediumN_* is the moles of N in WPM. The target moles of each element were calculated, and the formula was created, as described above. 

These three new formulations were then compared with commercially available basal salt formulations, WPM [8], MS [17], and DKW [23,24]. Eleven ‘core’ *Q. dumosa* culture lines that showed consistently positive growth of new shoot tips and comparatively low necrosis were chosen from one hundred and fifty-seven culture lines. Necrosis, for the purpose of data collection, was defined as the darkening of all stem tissue coupled with distortion to tissues, often coupled with excessive phenolic exudation (Figure 1c,e). Eight 1 cm shoot tip explants and eight 1 cm nodal segment explants with a single bud from each of the 11 ‘core’ culture lines were placed on the initial medium (Figure 2) with WPM salts or with one of the other five basal salt formulations for a total of 528 shoot tips and 528 nodal segments. All six media included WPM vitamins. Necrosis and all shoots 0.5 cm or greater were counted after 80 days of culture. The original shoot tip explant was not included in the count.

### 3.3. The Effect of Carbohydrate Source

After data from the basal salt study were analyzed, all culture lines were transferred to version 2 medium (Figure 2) for at least a month. Four additional lines began to show consistent positive growth of new shoots and decreased necrosis, leading them to produce enough shoots to be used in experiments. Eight shoot explants from the eleven core culture lines and the four ‘additional’ lines were each placed on version 2 medium with either the 30 g L^−1^ of sucrose or with sucrose substituted with either 30 g L^−1^ glucose (Sigma) or mannitol (Sigma) for a total of 360 shoot explants. Necrosis and shoots 0.25 cm or greater were counted after 80 days. 

### 3.4. The Effect of Activated Charcoal and Ascorbic Acid

After data from the carbohydrate study were analyzed, all culture lines were transferred to version 3 medium (Figure 2) for at least a month. The eleven core lines and nine additional lines, including the four used in the carbohydrate study, produced enough shoots to be used to test the effects of 5 g L^−1^ activated charcoal (AC) (Sigma) or 250 mg L^−1^ ascorbic acid (AA) (Phytotech) on culture growth. Six shoot tips from each of the eleven core culture lines and the nine additional lines were placed on a medium with 5 g L^−1^ AC, 250 mg L^−1^ AA, or no exogenous antioxidants for a total of 360 shoot explants. Necrosis and shoots with lengths 0.25 cm or greater were counted after 80 days. 

### 3.5. The Effect of Stress-Mitigating Additives 

After data from the carbohydrate study were analyzed, all culture lines were transferred to version 4 medium (Figure 2) for at least a month. The number of additional lines rose to sixteen, including the nine used in the antioxidant study. Of those, six had sufficient numbers of shoots to be used in only one of two concurrent experiments. In the first, six shoot tip explants from the eleven core lines and thirteen additional lines were placed on version 4 medium with 0, 1.3, 2.6, 5.2, or 10.5 mg L^−1^ sodium nitroprusside (SNP) (Sigma) for a total of 720 shoots. In the second experiment, six shoot tip explants from the eleven core lines and thirteen additional lines were placed on version 4 medium with 0, 51.5, or 103 mg L^−1^ γ-aminobutyric acid (GABA) (Sigma) for a total of 432 shoot tips. Necrosis and total shoots with lengths greater than 0.25 cm were counted after 80 days. 

### 3.6. The Effect of 6-Benzylaminopurine and Gibberellic Acid

After data from the carbohydrate study were analyzed, all culture lines were transferred to version 5 medium (Figure 2) for two months. Afterward, shoot tips from all 115 available culture lines, including the 11 core and 16 additional lines used previously, were placed on version 5 medium with 0, 0.33, 0.66, or 1 mg L^−1^ BAP and 0 or 0.5 mg L^−1^ gibberellic acid (GA_3_) (Phytotech). Four shoot tips from the eleven core and sixteen additional culture lines were placed on each of the eight treatments (864 shoots). The remaining 88 lines were divided in the following manner: culture lines with enough shoots contributed one shoot per treatment, and lines that had less than eight available shoots had their shoots randomly assigned to treatments without repeat. Necrosis, total shoots with lengths greater than 0.25 cm, and shoots greater than 2 cm were counted after 80 days of culture. 

### 3.7. Comparison between Initial and Final Medium Composition

After data from the BAP and gibberellic acid study were analyzed, all culture lines were transferred to the final *Q. dumosa* medium (Figure 2). Cultures were regularly subcultured onto a fresh medium monthly for seven months. Eight shoot explants from a total of twenty-five cultures lines consisting of the eleven core lines and fourteen additional lines were placed either on the initial medium consisting of WPM salts and vitamins, 30 g L^−1^ sucrose, 1 mg L^−1^ BAP, and 8 g L agar or the final medium consisting of Qd1 salts, WPM vitamins, 30 g L^−1^ glucose, 8 g L^−1^ agar, 250 mg L^−1^ ascorbic acid, 5.2 mg L^−1^ SNP, 103 mg L^−1^ GABA, and 0.66 mg L^−1^ BAP. Necrosis and total shoots with lengths greater than 0.25 cm were counted after 80 days. 

### 3.8. Rooting

After the analysis of the study on the effect of activated charcoal and ascorbic acid, but before the study on the effect of stress-mitigating additives, eight shoots from nine culture lines were rooted using three methods. In the incubation method, shoots were placed on a modified version 4 medium without BAP and 5 mg L^−1^ indole-3-butyric acid (IBA) for 30 days. In the pulse method, shoots were placed on a medium containing 20 mg L^−1^ IBA for 24 h and then transferred to a medium without plant growth regulators for 29 days. In the dip method, the cut ends of fresh shoots were placed in a 1 mg ml^−1^ IBA solution for 15 s and then placed on a medium without growth regulators for 30 days. For all three methods, the medium contained either 0 or 5 g L^−1^ AC to test its effects on root formation. In total, 432 shoot tips were used. The number of rooting shoots was counted after 30 days.

### 3.9. Statistical Analysis

The means of each growth response was taken for each individual culture line. Each treatment was presented as the average mean of all tested culture lines. All statistics were analyzed using R statistical software V 4.2.2 [55] with the following packages: ‘ggplot2’, ‘dplyr’, ‘psych’, ‘emmeans’, ‘lmtest’, ‘multcomview’, ‘plyr’, ‘car’, and ‘agricolae’. ANOVA and multi-factor ANOVA were used to determine possible interactions between growth response and a given treatment or interactions between treatments. When a significant effect or interaction was found, a Tukey post hoc test (*p* < 0.05) was used for multiple comparisons between treatments. Because many culture lines were used, data were also screened to track if a treatment was detrimental or beneficial for individual culture lines.

## 4. Conclusions

This medium, composed of Qd1 basal salts, woody plant medium vitamins, 30 g L^−1^ glucose, 8 g L^−1^ agar, 250 mg L^−1^ ascorbic acid, 5.2 mg L^−1^ SNP, 103 mg L^−1^ GABA, and 0.33–0.66 mg L^−1^ BAP, provided tissue culture conditions capable of keeping over a hundred *Q. dumosa* culture lines alive and, for the most part, growing consistently. The overall slow growth within most culture lines remains a bottleneck for producing enough material for larger and more comprehensive tests. The low rooting rates and slow growth of rooted plants require further investigation because the goal of using tissue culture for ex situ conservation is to eventually return plants to soil. Such experiments are now possible due to the step-by-step improvements to the *Q. dumosa* culture medium, which initially showed high rates of stress and necrosis and developed into a medium capable of producing a sustained population growth of healthy shoots. 

## Figures and Tables

**Figure 1 plants-13-01148-f001:**
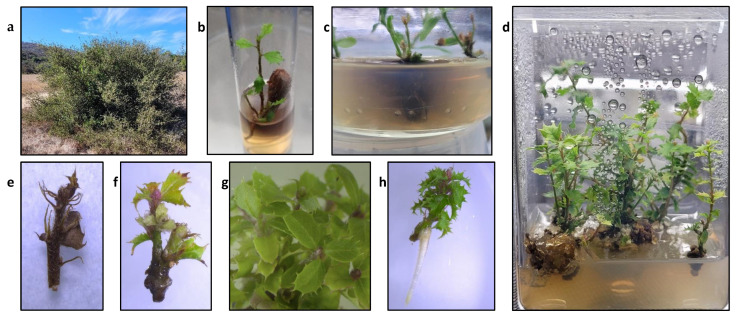
*Quercus dumosa* in nature and in tissue culture. (**a**) A wild *Quercus dumosa* growing in situ. (**b**) A *Quercus dumosa* acorn germinating in vitro, which would be sliced into nodal segments to create a culture line. (**c**) Phenolic exudation from a culture. (**d**) *Quercus dumosa* shoot cultures growing on medium. Considerable callus growth is present where the shoots make contact with the medium. (**e**) A necrotic shoot tip with darkened tissues and a withered appearance. (**f**) A living shoot tip with axillary bud growth but displaying leaf discoloration and other signs of stress after a month on WPM medium. (**g**) A cluster of oak shoots growing from a single shoot tip grown on medium with Qd1 salts. (**h**) A rooted shoot tip.

**Figure 2 plants-13-01148-f002:**
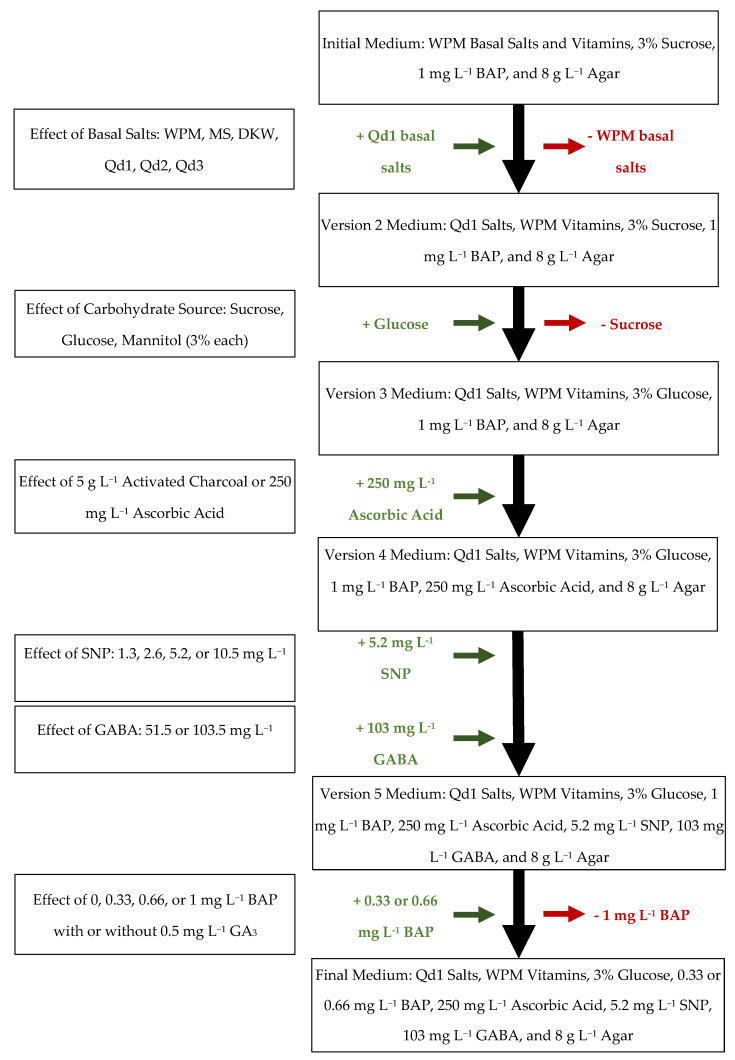
The development of a tissue culture medium capable of growing multiple culture lines of *Quercus dumosa*.

**Table 1 plants-13-01148-t001:** Element composition of *Quercus dumosa* shoot tips and first leaves.

Element	% of Total Mass
N	1.41
K	1.2
Ca	0.45
Mg	0.17
P	0.16
S	0.080
Mn	0.038
Fe	0.00386
B	0.0022
Zn	0.0022
Cu	0.00089

**Table 2 plants-13-01148-t002:** Composition in milligrams per liter of three widely used commercial basal salt mixes and three basal salt mixes developed for *Quercus dumosa* based on the elemental composition of shoot tips and first leaves.

Compound	WPM ^1^	MS ^2^	DKW ^3^	Qd1 ^4^	Qd2 ^5^	Qd3 ^6^
NH_4_NO_3_	400	1650	1416	240	431	50
H_3_BO_3_	6.2	6.2	4.8	1.89	4.11	1.42
CaCl_2_	72.5	332	112.5	60	0	250
Ca(NO_3_)_2_•4H_2_O	386	0	1367	276.5	603	402.5
CuSO_4_•5H_2_O	0.25	0.025	0.25	0.527	1.15	2.33
Na_2_EDTA•2H_2_O	37.3	37.3	45.4	37.3	37.3	37.3
FeSO_4_•7H_2_O	27.85	27.8	33.8	2.88	6.28	11.2
MgSO_4_	180.7	180.7	361.5	0	0	0
MnSO_4_•H_2_O	22.3	16.9	33.5	0	0	0
Na_2_MoO_4_•2H_2_O	0.25	0.25	0.39	0.25	0.25	0.25
KH_2_PO_4_	170	170	265	0	0	227
K_2_SO_4_	990	0	1559	62.5	136	132.7
ZnSO_4_•7H_2_O	8.6	8.6	17	1.42	3.09	6.45
KNO_3_	0	1900	0	393	860	505
KI	0	0.83	0	0	0	0
KCl	0	0	0	0	0	322
NH_4_H_2_PO_4_	0	0	0	0.089	0.195	0
MnCl_2_	0	0	0	0.013	0.028	49.02
Mg(NO_3_)_2_•6 H_2_O	0	0	0	0.269	0.59	455
NiSO_4_•6H_2_O	0	0	0.005	0	0	0
CoCl_2_•6H_2_O	0	0.025	0	0	0	0

^1^ Formula found in McCown and Lloyd [8]. ^2^ Formula found in Murashige and Skoog [17]. ^3^ Formula found in Driver and Kuniyuki [23] and McGranahan et al. [24] ^4^ Based on mineral composition of *Quercus dumosa* shoot tips and first leaf using the calculation methods found in Monteiro et al. [12] ^5^ Based on mineral composition of *Quercus dumosa* shoot tips and first leaf using the calculation methods found in Terrer and Tomás [14] ^6^ Based on mineral composition of *Quercus dumosa* shoot tips and first leaf using the calculation methods found in Staikidou et al. [16].

**Table 3 plants-13-01148-t003:** Effect of basal salt mixture on growth response of shoot tip and nodal segment explants from 11 ‘core ^1^’ *Quercus dumosa* culture lines ^2^.

Basal Salt Mixture ^3^	Necrosis (% of Explants)	# Average Shoots Per Explant ^4^	Shoots > 1 cm in Length (%)
**Shoot Tip Explants**			
WPM	15.6 ± 3.2 A ^5^	1.0 ± 0.1 A	36.6 ± 4.6 AB
DKW	41.8 ± 7.1 B	0.5 ± 0.1 C	14.8 ± 5.6 AB
MS	45.2 ± 5.5 B	0.5 ± 01 BC	10.1 ± 3.1 B
Qd1	2.1 ± 1.6 A	1.1 ± 0.1 A	41.2 ± 6.2 A
Qd2	4.0 ± 1.8 A	0.9 ± 0.2 A	28.8 ± 9.1 AB
Qd3	1.9 ± 1.3 A	0.9 ± 0.1 AB	21.3 ± 8.0 AB
**Nodal Segment**			
WPM	39.6 ± 7.3 AB	0.78 ± 0.1 A	31.3 ± 6.3 A
DKW	55.1 ± 6.2 B	0.21 ± 0.1 B	9.7 ± 4.1 AB
MS	56.9 ± 8.7 B	0.17 ± 0.1 B	4.8 ± 2.1 B
Qd1	20.5 ± 6.0 A	0.6 ± 0.2 AB	24.1 ± 8.6 AB
Qd2	17.7 ± 7.1 A	0.4 ± 0.1 AB	16.0 ± 6.6 AB
Qd3	21.2 ± 6.5 A	0.5 ± 0.1 AB	11.3 ± 8.0 AB

^1^ Eleven of an initial one hundred and fifty-seven lines that showed consistent positive growth and had sufficient numbers of shoots and nodal segments. ^2^ Eight shoot tips or nodal segment explant per treatment per culture line were used. The average of those eight were then pooled with the results of the other culture lines to compare between treatments (*N* = 11). A total of 48 of each tissue type were used per culture line for a total of 528 total across all 11 culture lines. ^3^ Additional medium constituents include: WPM vitamins, 30 g L^−1^ sucrose, 1 mg L^−1^ BAP, and 8 g L^−1^ agar. ^4^ Shoots 0.5 cm or greater alone were counted. The original explant’s shoot was not included in the count. ^5^ Capitalized Tukey post hoc letters (*p* < 0.05) signify differences between treatments.

**Table 4 plants-13-01148-t004:** Effect of carbohydrate source on growth of the 11 core *Quercus dumosa* culture lines and 4 additional lines ^1,2^.

Carbohydrate Source ^3^	Culture Line Type	Necrosis (% of Explants)	# Average Shoots Per Explant ^4^
Sucrose	15 Total Lines	9.6 ± 1.3 B ^5^	1.6 ± 0.2 B
	11 Core Lines ^6^	4.4 ± 2.8 ab	1.69 ± 0.2 ab
	4 Additional Lines	24.0 ± 6.7 b	1.19 ± 0.1 b
Glucose	15 Total Lines	1.8 ± 3.2 A	1.89 ± 0.2 A
	11 Core Lines	0.4 ± 0.7 a	2.06 ± 0.1 a
	4 Additional Lines	5.6 ± 3.9 ab	1.40 ± 0.1 b
Mannitol	15 Total Lines	64.1 ± 10.7 C	0.39 ± 0.1 C
	11 Core Lines	51.0 ± 12.6 c	0.52 ± 0.1 c
	4 Additional Lines	100 ± 0.0 d	0.00 ± 0.0 c

^1^ Additional lines refer to lines added to experiments in addition to the 11 core lines used in the basal salt study (Table 3). ^2^ Eight shoot tips per treatment per culture line were used. The average of those eight were then pooled with the results of the other culture lines to compare between treatments (*N* = 15). A total of 24 shoot tips were used per culture line for a total of 360 across all 15 culture lines. ^3^ 30 g L^−1^ carbohydrate. Additional medium constituents include: Qd1 basal salts (Table 2), woody plant medium vitamins, 1 mg L^−1^ BAP, and 8 g L^−1^ agar. ^4^ All shoots greater than 0.25 cm were counted in contrast to Table 5, where shoots only 0.5 cm or greater were counted. The original shoot tip explant was not included in the count. ^5^ Capitalized Tukey post hoc letters (*p* < 0.05) signify differences between treatments across all tested lines, whereas lowercase letters are between treatments and line type. ^6^ Data are presented as both total lines, as well as divided into the 11 core lines and additional lines because the 11 core lines are present in every experiment, and so is a baseline for evaluating *Q. dumosa* culture medium development over time.

**Table 5 plants-13-01148-t005:** Effect of antioxidants on growth of the 11 core *Quercus dumosa* culture lines and 9 additional slow-growing lines ^1^.

Treatment	Culture Line Type	Necrosis (% of Explants)	# Average Shoots per Explant
Version 3 Medium ^2^	20 Total Lines	9.6 ± 3.4 B ^3^	1.6 ± 0.1 A
	11 Core Lines ^4^	0.0 ± 0.0 a	1.8 ± 0.2 a
	9 Additional Lines	21.3 ± 5.6 b	1.2 ± 0.1 bc
+5 g L^−1^ Activated Charcoal	20 Total Lines	0.0 ± 0.0 A	1.1 ± 0.1 B
	11 Core Lines	0.0 ± 0.0 a	1.1 ± 0.2 bc
	9 Additional Lines	0.0 ± 0.0 a	1.0 ± 0.0 c
+250 mg L^−1^ Ascorbic Acid	20 Total Lines	1.3 ± 0.9 A	1.7 ± 0.1 A
	11 Core Lines	0.0 ± 0.0 a	1.9 ± 0.2 a
	9 Additional Lines	2.8 ± 2.1 a	1.4 ± 0.1 b

^1^ Six shoot tips per treatment per culture line were used. The average of those six were then pooled with the results of the other culture lines to compare between treatments (*N* = 20). A total of 18 shoot tips were used per culture line for a total of 360 across all 20 culture lines. ^2^ Medium constituents include: Qd1 basal salts (Table 2), woody plant medium vitamins, 30 g L^−1^ glucose (Table 4), 1 mg L^−1^ BAP, and 8 g L^−1^ agar. ^3^ Capitalized Tukey post hoc letters (*p* < 0.05) signify differences between treatments across all tested lines, whereas lowercase letters are between treatments and line type. ^4^ Data are presented as both total lines, as well as divided into the 11 core lines and additional lines because the 11 core lines are present in every experiment, and so is a baseline for evaluating *Q. dumosa* culture medium development over time.

**Table 6 plants-13-01148-t006:** Effect of sodium nitroprusside (SNP) on the growth of 11 core and 13 additional *Quercus dumosa* culture lines ^1^.

Treatment	Culture Line Type	Necrosis (% of Explants)	# Average Shoots per Explant
Version 4 Medium ^2^	24 Total lines	7.3 ± 2.4 B ^3^	1.8 ± 0.1 B
	11 Core lines ^4^	2.6 ± 2.3 a	2.2 ± 0.2 abcd
	13 Additional lines	11.5 ± 3.6 a	1.5 ± 0.1 d
+1.3 mg L^−1^ SNP	24 Total Lines	4.2 ± 1.9 AB	1.8 ± 0.2 AB
	11 Core Lines	0.0 ± 0.0 a	2.3 ± 0.2 bcd
	13 Additional Lines	7.7 ± 3.3 a	1.5 ± 0.2 d
+2.6 mg L^−1^ SNP	24 Total Lines	4.2 ± 2.5 AB	1.9 ± 0.1 AB
	11 Core Lines	0.0 ± 0.0 a	2.4 ± 0.2 ab
	13 Additional Lines	7.7 ± 4.4 a	1.6 ± 0.1 cd
+5.2 mg L^−1^ SNP	24 Total Lines	1.0 ± 1.0 A	2.0 ± 0.1 A
	11 Core Lines	0.0 ± 0.0 a	2.3 ± 0.1 abc
	13 Additional Lines	1.9 ± 1.9 a	1.8 ± 0.1 bcd
+10.5 mg L^−1^ SNP	24 Total Lines	1.0 ± 1.0 A	2.0 ± 0.1 AB
	11 Core Lines	0.0 ± 0.0 a	2.5 ± 0.2 a
	13 Additional Lines	1.9 ± 1.9 a	1.7 ± 0.1 abcd

^1^ Six shoot tips per treatment per culture line were used. The average of those six was then pooled with the results of the other culture lines to compare between treatments (*N* = 24). A total of 30 shoot tips were used per culture line for a total of 720 across all 24 culture lines. ^2^ Medium consists of Qd1 basal salts (Table 2), WPM vitamins, 30 g L^−1^ glucose (Table 4), 1 mg L^−1^ BAP, 8 g L^−1^ agar, and 250 mg L^−1^ ascorbic acid (Table 5). ^3^ Capitalized Tukey post hoc letters (*p* < 0.05) signify differences between treatments across all tested lines, whereas lowercase letters are between treatments and line type. ^4^ Data are presented as both total lines, as well as divided into the 11 core lines and additional lines because the 11 core lines are present in every experiment, and so is a baseline for evaluating *Q. dumosa* culture medium development over time.

**Table 7 plants-13-01148-t007:** The effect of γ-aminobutyric acid (GABA) on the growth of 11 core and 13 additional *Quercus dumosa* culture lines ^1^.

Treatment	Culture Line Type	Necrosis (% of Explants)	# Average Shoots Per Explant
Version 4 Medium ^2^	24 Total Lines	2.1 ± 1.2 A ^3^	2.0 ± 0.1 B ^3^
	11 Core Lines ^4^	1.5 ± 1.5 a	2.6 ± 0.2 a
	13 Additional Lines	2.6 ± 1.7 a	1.5 ± 0.1 b
+51.5 mg L^−1^ GABA	24 Total Lines	3.5 ± 1.7 A	2.0 ± 0.1 B
	11 Core Lines	0.0 ± 0.0 a	2.5 ± 0.2 a
	13 Additional Lines	6.4 ± 3.0 a	1.6 ± 0.1 b
+103 mg L^−1^ GABA	24 Total Lines	1.4 ± 1.0 A	2.2 ± 0.1 A
	11 Core Lines	0.0 ± 0.0 a	2.6 ± 0.1 a
	13 Additional Lines	2.6 ± 1.7 a	1.8 ± 0.1 b

^1^ Six shoot tips per treatment per culture line were used. The average of those six was then pooled with the results of the other culture lines to compare between treatments (*N* = 24). A total of 18 shoot tips were used per culture line for a total of 432 across all 24 culture lines. ^2^ Medium consists of Qd1 basal salts (Table 2), WPM vitamins, 30 g L^−1^ glucose (Table 4), 1 mg L^−1^ BAP, 8 g L^−1^ agar, and 250 mg L^−1^ ascorbic acid (Table 5). ^3^ Capitalized Tukey post hoc letters (*p* < 0.05) signify differences between treatments across all tested lines, whereas lowercase letters are between treatments and line type. ^4^ Data are presented as both total lines, as well as divided into the 11 core lines and additional lines because the 11 core lines are present in every experiment, and so is a baseline for evaluating *Q. dumosa* culture medium development over time.

**Table 8 plants-13-01148-t008:** The effect of 6-benzylaminipurine (BAP) on the growth of 11 core, 16 additional, and 88 static ^1^ *Quercus dumosa* culture lines ^2^.

BAP (mg L^−1^) ^3^	Culture Type	Necrosis (% of Explants)	# Average Shoots Per Explant	Shoots Greater Than 2 cm (%)
1	94 Total Lines	10.5 ± 3.0 A ^4^	1.8 ± 0.1 A	14.4 ± 3.4 AB
	11 Core Lines ^5^	0.0 ± 0.0 a	2.6 ± 0.2 ab	31.3 ± 9.1 ab
	16 Additional Lines	4.2 ± 2.5 a	1.8 ± 0.2 abcd	11.7 ± 6.5 bc
	67 Static Lines	16.5 ± 4.5 a	1.6 ± 0.2 cd	10.7 ± 4.3 bc
0.66	95 Total Lines	9.0 ± 2.9 A	1.9 ± 0.1 A	19.0 ± 4.1 A
	11 Core Lines	0.0 ± 0.0 a	2.6 ± 0.2 a	33.8 ± 7.0 ab
	16 Additional Lines	1.6 ± 1.5 a	1.9 ± 0.2 abcd	17.2 ± 8.0 bc
	68 Static Lines	15.1 ± 4.4 a	1.8 ± 0.2 bcd	15.4 ± 5.7 bc
0.33	95 Total Lines	10.1 ± 2.9 A	1.7 ± 0.1 A	21.2 ± 4.2 A
	11 Core Lines	0.0 ± 0.0 a	2.3 ± 0.2 abc	49.4 ± 15.0 a
	16 Additional Lines	3.4 ± 2.3 a	1.6 ± 0.2 cd	23.7 ± 8.7 abc
	68 Static Lines	16.2 ± 4.4 a	1.6 ± 0.1 cd	11.8 ± 4.3 bc
0	94 Total Lines	25.1 ± 4.1 B	0.9 ± 0.1 B	4.1 ± 1.7 B
	11 Core Lines	7.5 ± 2.5 a	1.1 ± 0.1 e	6.3 ± 9.1 bc
	16 Additional Lines	9.6 ± 3.9 a	1.1 ± 0.1 de	6.5 ± 6.5 bc
	67 Static Lines	37.7 ± 5.9 b	0.7 ± 0.1 e	2.2 ± 4.3 c

^1^ Lines that produced enough shoots to maintain their numbers but otherwise did not produce enough shoots for use in prior experiments. ^2^ Four shoot tips per treatment per core and additional culture line were used. Many lines had insufficient numbers of shoots for each treatment. Shoot tips from static lines were distributed as equally as possible with numbers ranging from two to zero shoots per treatment, resulting in 67 or 68 total lines for each BAP concentration from the total of 88 static lines. *N* = 94 for 0 or 1 mg L^−1^ BAP or 95 for 0.33 or 0.66 mg L^−1^ BAP. ^3^ Medium constituents include: Qd1 basal salts (Table 2), WPM vitamins, 30 g L^−1^ glucose (Table 4), 8 g L^−1^ agar, 250 mg L^−1^ ascorbic acid (Table 5), 5.2 mg L^−1^ SNP (Table 6), and 103 mg L^−1^ GABA (Table 7). ^4^ Capitalized Tukey post hoc letters (*p* < 0.05) signify differences between treatments across all tested lines, whereas lowercase letters are between treatments and line type. ^5^ Data are presented as both total lines, as well as divided into the 11 core lines and additional lines because the 11 core lines are present in every experiment, and so is a baseline for evaluating *Q. dumosa* culture medium development over time.

**Table 9 plants-13-01148-t009:** Comparison between initial ^1^ and final ^2^ medium formulations on the growth of 11 core and 14 additional *Quercus dumosa* culture lines ^3^.

Medium Type	Culture Line Type	Necrosis (% of Explants)	# Average Shoots Per Explant
Initial ^1^	25 Total Lines ^3^	51.0 ± 6.9 B ^4^	0.9 ± 0.2 B
	11 Core Lines ^5^	29.5 ± 8.1 b	1.3 ± 0.3 b
	14 Additional Lines	67.9 ± 7.8 c	0.5 ± 0.1 b
Final ^2^	25 Total Lines	6.5 ± 1.9 A	1.9 ± 0.2 A
	11 Core Lines	1.1 ± 1.1 a	2.4 ± 0.4 a
	14 Additional Lines	10.7 ± 2.8 ab	1.5 ± 0.1 ab

^1^ Woody plant medium and vitamins (see footnote 3), 30 g L^−1^ sucrose, 1 mg L^−1^ BAP, and 8 g L^−1^ agar. ^2^ Medium constituents include: Qd1 basal salts (Table 2), WPM vitamins, 30 g L^−1^ glucose (Table 4), 8 g L^−1^ agar, 250 mg L^−1^ ascorbic acid (Table 5), 5.2 mg L^−1^ SNP (Table 6), 103 mg L^−1^ GABA (Table 7), and 0.66 mg L^−1^ BAP. ^3^ Eight shoot tips per treatment per culture line were used. The average of those eight were then pooled with the results of the other culture lines to compare between treatments (*N* = 25). A total of 16 shoot tips were used per culture line for a total of 400 across all 25 culture lines. ^4^ Capitalized Tukey post hoc letters (*p <* 0.05) signify differences between treatments across all tested lines, whereas lowercase letters are between treatments and line type. ^5^ Data are presented as both total lines, as well as divided into the 11 core lines and additional lines because the 11 core lines are present in every experiment, and so is a baseline for evaluating *Q. dumosa* culture medium development over time.

**Table 10 plants-13-01148-t010:** Effect of rooting method and activated charcoal on 9 *Quercus dumosa* culture lines ^1^.

Method	Activated Charcoal	Average Rooting (%)	# of Culture Lines with at Least One Rooted Plant	Range of Rooting ^2^ (%)
Long Incubation ^3^	5 g	4.2 ± 2.1 C	3	12.5–12.5
One-Day Pulse ^4^		47.2 ± 10.2 A ^6^	8	25–100
15-Second Dip ^5^		0.0 ± 0.0 ^7^	0	N/A
Long Incubation	0	16.7 ± 6.9 BC	5	12.5–62.5
One-Day Pulse		38.9 ± 10.1 AB	7	12.5–75
15-Second Dip		0.0 ± 0.0 ^7^	0	N/A

^1^ Eight shoot tips per treatment per culture line were used. The average of those eight were then pooled with the results of the other culture lines to compare between treatments (*N* = 9). A total of 48 shoot tips were used per culture line for a total of 432 across all 9 culture lines. ^2^ Only culture lines that produced at least one root were included in this range. ^3^ Incubation in 5 mg L^−1^ IBA for 30 days onto BAP-free version 4 medium (see Table 2) with or without 5 g L^−1^ activated charcoal. ^4^ 24-h incubation in medium with 20 mg L^−1^ IBA before transfer to version 4 medium with or without AC 5 g L^−1^ activated charcoal for 29 days. ^5^ A 15 s dip into 1 mg mL^−1^ IBA before transferring to version 4 with or without AC 5 g L^−1^ activated charcoal for 30 days. ^6^ Capitalized Tukey post hoc letters (*p <* 0.05) signify differences between treatments. ^7^ Excluded from analysis because no roots resulted from the dip method.

## Data Availability

The original contributions presented in the study are included in the article; further inquiries can be directed to the corresponding author.

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
