# Peer review of "Development of a Micropropagation Protocol for the Ex Situ Conservation of Nuttall’s Scrub Oak (Quercus dumosa)"

_plants, 2024, doi:10.3390/plants13081148_

Round 1
Reviewer 1 Report
Comments and Suggestions for Authors
Reviewer 2 Report
Comments and Suggestions for Authors
Dear authors
The importance of this study in very clear: showing the problems and difficulties in front of a anyone who needs to develop a new plant medium especially when many genome types are involved. This is the strong side of the presented study. And that is why a major revision of the MS is needed. The stages of developing the composition of the new medium is important part of the results. The second problem is the generalization of the results, since you have dealt with many genomes. Therefore, I suggest to show the results of the main treatments only, and discuss the different responses of the plant lines in a separate paragraph. Also pay attention to the statistical differences on one hand and the significant/insignificant difference when the statistical test shows a significant difference on the other hand.
Other specific comments are seen on the MS

Reviewer 3 Report
Comments and Suggestions for Authors
The manuscript reviewed deals with the design of an improved protocol for establishing in vitro as many lines of Quercus dumosa as necessary to maintain the genetic diversity of this endangered species. A new medium was developed, based on plant minerals analysis, and various additives were successfully tested.
The experiments are well-designed, and the extent of the study is quite impressive. The initial goal is reached, with the improved medium allowing the successful culture of some one hundred different lines.
I particularly appreciated the fact that the study was aimed at preserving the actual genetic diversity of the species, and not at the establishment of a few in vitro clones.
I have no major concerns about the text; I feel the DK medium is not enough commented compared to WPM et MS, although it was specifically developed for a Quercus species. Also, the authors do not comment on the relative ionic strength of the various media used or developed (it seems indeed that Qd1 has a lower overall concentration in minerals). This can be easily shown in an additional table, which I believe would be an improvement to the manuscript. In the same way, I was wondering whether, even if WPM is indeed lower in minerals than MS, seeds would benefit from germinating on an even less concentrated substrate; moreover, germination on too concentrated a medium might impose an early stress on plantlets, and consequently on explants derived from them, with unknown consequences on their future behaviour. Since seeds contain all necessary elements for germination, and will germinate better in low osmotic pressure conditions, we routinely germinate seeds on a medium containing only ¼ MS macro elements with 10-15g sucrose.
Other minor remarks are as follows:
· In a number of experiments, some chemicals are added to the media; as some of these can be added before or after autoclaving, please provide the reader with more information on whether glucose (for example), ascorbic acid, SNP, gibberellic acid, and GABA were co-autoclaved or filter-sterilized.
· As is, Tables 1 and 2 are difficult to read for lack of a logical ranking. Please provide new tables following the generally accepted ranking, i.e. from the most abundant chemical to the least concentrated one. Also, in general, do not include in the captions too many unnecessary details.
· Overall, the manuscript is lengthy, with a number of sections that could be suitably shortened without any loss of meaning. A few examples are provided below, but no editing in full has been attempted.
o Lines 715-782: much redundancy in the first lines, just state that “Qd1,Qd2,Qd3 were developed as follows: ” and then start again at line 719; the transcription of mass into mole is sensible, but does not require such extensive explanations as readers of “Plants” can be expected to be familiar with this conversion. (also in line 731, use “respectively” instead of “respectfully”).
o References to B. gasipaes do not relate to the main core of the research and should probably be removed altogether (besides, if natural diversity preservation is the main goal, introducing somatic embryogenesis, through which somaclonal variation cannot be excluded, is probably not advisable; see lines 544-544)
o Some sentences could be better contrived, e.g. lines 394-395; line 477; line 658; lines 676-677; etc
In conclusion, this manuscript describes a very nice piece of research and should be published. Overall,
· The research described is quite impressive, and the goals initially set are nearly all achieved;
· The manuscript can be improved on minor points (see above)
· Finally, the manuscript should be shortened by carefully eliminating overlong or redundant sentences.
Comments on the Quality of English LanguageThe text should be edited for concision.
Author Response
Thank you very much for taking the time to read through this manuscript. Time is precious, so I truly value your input. I've included responses to each comment in the attached .docx file.

Reviewer 4 Report
Comments and Suggestions for Authors
The work is relevant and helpful for those working the field of plant multiplication, particularly in endangered woody species. The manuscript can be accepted after minor language editing.
Comments on the Quality of English LanguageThe English language needs minor editing: line 671 "a protocol less efficient" should be replaced with "a less efficient protocol".
Author Response
Thank you very much for taking the time to review this manuscript. Time is precious, so I truly value your input.
I edited several sentences throughout the manuscript in addition to the example, such as the sentences in lines 394-395; 477; 656; and 676-677, with some redundant sentences deleted from the methods section.
Your example made me see that I did not communicate what I hoped to do. I instead made the following change instead of your suggestion.
From: "Q. dumosa grows slowly in vitro, so these significant losses in tissue made developing a protocol less efficient"
To: "Q. dumosa grows slowly in vitro, so these significant losses in tissue represent a considerable delay in protocol development"
Reviewer 5 Report
Comments and Suggestions for Authors
Joseph and his colleagues aim to develop a method of cultivating many Quercus dumosa culture lines in vitro for the Ex situ Conservation of endangered oak species and a new basal medium. The manuscript enriches our knowledge about the Ex situ Conservation of endangered oak species. The conclusions are supported by the data, and the submitted manuscript is written clearly and interest to the readers. I suggest to accepting the manuscript in present form.
Author Response
Thank you for taking the time to read this manuscript. Time is precious, so I truly value your input. I also appreciate your kind words; this manuscript represents a lot of work. I hope it may be of use not only to others working with oaks, but to anyone hoping to use tissue culture for conservation regardless of species.